# Thermodynamic Modeling of the Ag-Cu-Sn Ternary System

**Qingsong Tong, Jing Ge, Maohua Rong *** , **Jielong Li, Jian Jiao, Lu Zhang and Jiang Wang ***

School of Materials Science and Engineering & Guangxi Key Laboratory of Information Materials, Guilin University of Electronic Technology, Guilin 541004, China

***** Correspondence: rongmh124@guet.edu.cn (M.R.); waj124@guet.edu.cn (J.W.)

**Abstract:** In this work, combined with previous assessments of the Ag-Cu, Ag-Sn and Cu-Sn binary systems, thermodynamic modeling of the Ag-Cu-Sn ternary system was performed using the CALPHAD method using the reported phase diagram data and thermodynamic data. The solution phases including Liquid, fcc, bcc, hcp, bct(Sn) and diamond(Sn) were modeled as substitutional solutions and their excess Gibbs energies were expressed by the Redlich–Kister–Muggianu polynomial. The solubility of the third element in binary intermetallic compounds was not taken into account due to the fact that ternary solubilities for most of the binary compounds are not significant. Thermodynamic properties of liquid alloys, liquidus projection, several vertical sections and isothermal sections were calculated, which were in reasonable agreement with the reported experimental data. Finally, a set of self-consistent thermodynamic parameters formulating the Gibbs energies of various phases in the Ag-Cu-Sn ternary system was obtained.

**Keywords:** solders and brazing alloys; Ag-Cu-Sn; phase equilibria; CALPHAD

## 1. Introduction

Sn-Ag-Cu alloys have been developed as the most promising lead-free solders because Sn-Pb solders have been restricted in the microelectronics industry due to the toxicity of Pb [1–4]. Additionally, Ag-Cu-Sn-based brazing alloys have been used in industrial applications [5,6]. To reduce the costs of the solders and brazing alloys, some alloying elements such as In, Zn, Sb, Bi, Ni and rare earth metals are added to Sn-Ag-Cu alloys and Ag-Cu-Sn-based alloys [7–16]. The phase equilibria and thermodynamic information of Ag-Cu-Sn-based alloys are essential to understand the role of alloying elements and to design new Sn-Ag-Cu-based lead-free solders and Ag-Cu-Sn-based brazing alloys [17]. As an essential contribution to develop a consistent and available thermodynamic database of the multicomponent Ag-Cu-Sn-based alloys, the present work thermodynamically modeled the Ag-Cu-Sn ternary system.

Experimental studies and thermodynamic calculations of phase equilibria of the Ag-Cu-Sn ternary system were performed [18–29]. Gebhardt et al. [18] systematically conducted the experimental study of the Ag-Cu-Sn ternary system. Miller et al. [19], Loomans et al. [20], Moon et al. [21], Lewis et al. [22] and Park et al. [23] experimentally investigated the eutectic reaction of the Ag-Cu-Sn ternary system. Several isothermal sections and vertical sections of the Ag-Cu-Sn ternary system were determined by Ohnuma et al. [24], Yen et al. [25], Marjanovic et al. [26] and Fima et al. [27]. Based on the experimental results, the Ag-Cu-Sn ternary system was calculated by Moon et al. [21], Ohnuma et al. [24], Yen et al. [25], Gierlotka et al. [28] and the unpublished COST 531 Database for Lead-free Solders [29]. Recently, Luef et al. [30] experimentally determined the mixing enthalpy of the liquid phase in the Ag-Cu-Sn ternary system, and Kopyto et al. [31] experimentally determined the activity of Sn in the liquid phase, which was not considered in these thermodynamic calculations of the Ag-Cu-Sn ternary system reported by [21,24,25]. In this work, the previous assessments of the Ag-Cu [32], Ag-Sn [33] and Cu-Sn [34] binary systems were reviewed, and then thermodynamic modeling of the Ag-Cu-Sn ternary system

was performed using the CALPHAD method based on the reported phase equilibria and thermodynamic data.

## 2. Literature Information

### 2.1. *The Ag-Cu Binary System*

Thermodynamic calculations of the Ag-Cu binary system were reported [32,35–38]. The calculated results by He et al. [32] and Witusiewicz et al. [38] are in good agreement with the experimental results. Experimental results [39–45] show that the mixing enthalpy of liquid Ag-Cu alloys is dependent on temperature. Considering the temperature dependence of the mixing enthalpy of liquid Ag-Cu alloys, He et al. [32] re-optimized the Ag-Cu binary system in better agreement with the experimental data [46,47]. Therefore, the calculated results by He et al. [32] were used in this work. The calculated Ag-Cu binary phase diagram is shown in Figure 1a.

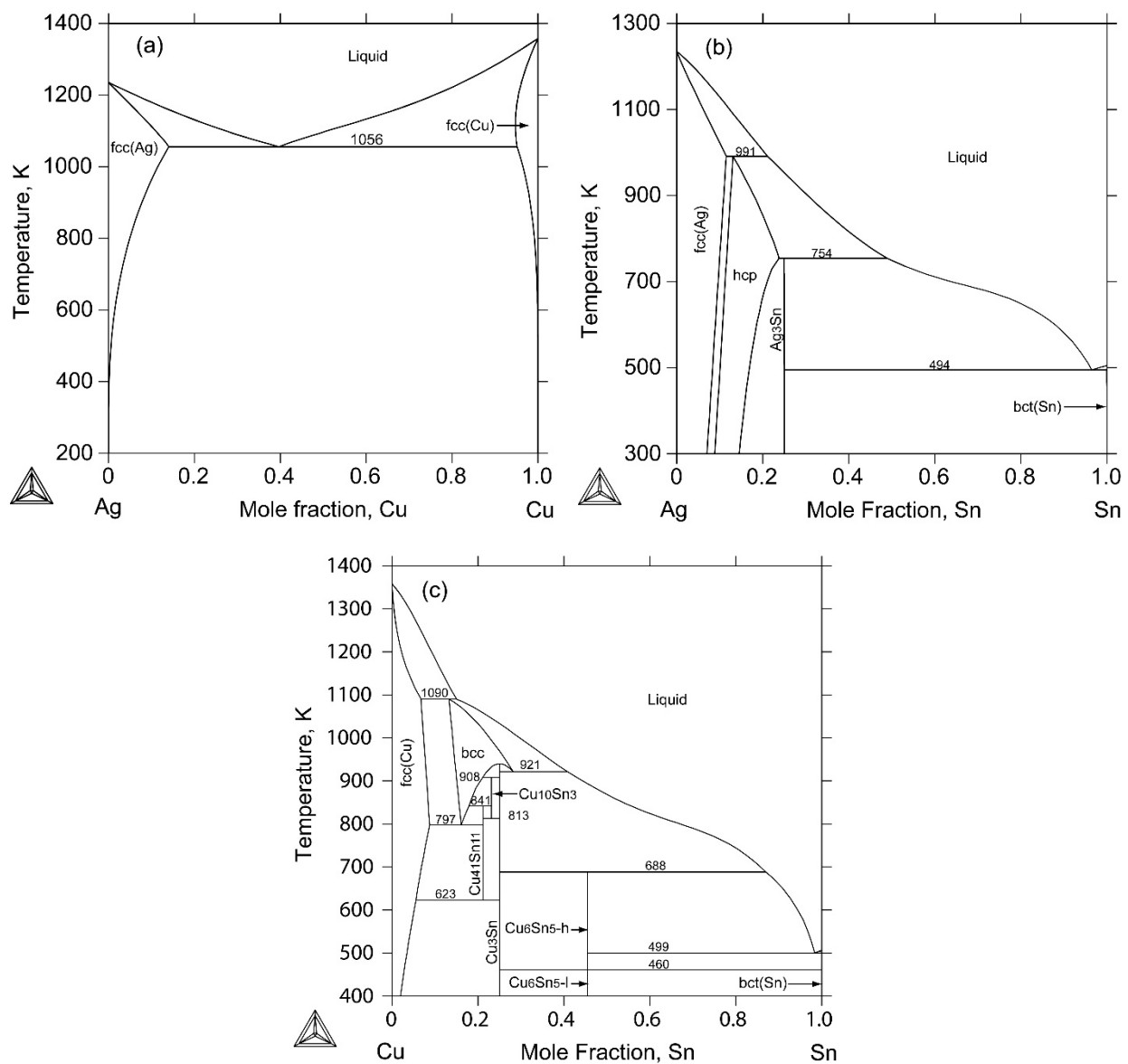

**Figure 1.** Calculated (**a**) Ag-Cu, (**b**) Ag-Sn and (**c**) Cu-Sn sub-binary phase diagrams by He et al. [32], Du et al. [33] and Liu et al. [34].

## 2.2. The Ag-Sn Binary System

Thermodynamic assessments of the Ag-Sn binary system were reported [33,48–52]. The intermetallic compound $Ag_3Sn$ was described as stoichiometric in the assessment of the Ag-Sn binary system reported by Chevalier et al. [49]. Kattner et al. [50] assessed the Ag-Sn binary system including the phase transition of β-Sn ↔ α-Sn. Although the calculated results by Oh et al. [51] are in general agreement with the experimental data, the calculated eutectic temperature is lower than the experimental data [53–55]. Moon et al. [21] re-assessed the stability of the intermetallic compound $Ag_3Sn$ to reproduce the experimental results of the eutectic reaction. Wang et al. [52] re-optimized thermodynamic parameters of the hcp phase in the Ag-Sn binary system using the SGTE (Scientific Group Thermodata Europe) database reported by Dinsdale [56]. The Gibbs energy expression of the liquid phase in the Ag-Sn binary system was re-optimized by Du et al. [33] considering the mixing enthalpy of liquid alloys measured by Flandorfer et al. [57]. The calculated results by Du et al. [33] are consistent with the phase diagram data [53–55] and thermodynamic data [57], and thus were used in this work. Figure 1b shows the calculated Ag-Sn binary phase diagram by Du et al. [33].

## 2.3. The Cu-Sn Binary System

Thermodynamic calculations of the Cu-Sn binary system were performed by Moon et al. [21], Shim et al. [58], Gierlotka et al. [59], Wang et al. [60], Miettinen [61], Li et al. [62], Liu et al. [63], Li et al. [64] and Liu et al. [34]. The different thermodynamic models were used to describe the bcc-A2 and $D0_3$ phases (ordered bcc-A2 phase). Moon et al. [21], Shim et al. [58] and Gierlotka et al. [59] using the substitutional solution model and Wang et al. [60] using the two-sublattice model $((Cu,Sn)_{0.75}(Cu,Sn)_{0.25})$ described the bcc-A2 and $D0_3$ phases. Meanwhile, Miettinen [61] and Li et al. [62] described the bcc-A2 and $D0_3$ phases using the substitutional solution model. Liu et al. [63] described the bcc-A2 and $D0_3$ phases using the two-sublattice model $((Cu,Sn)_{0.5}(Cu,Sn)_{0.5})$ to describe the A2→B2→$D0_3$ second-order transition, while Li et al. [64] used the four-sublattice model $((Cu,Sn)_{0.25}(Cu,Sn)_{0.25}(Cu,Sn)_{0.25}(Cu,Sn)_{0.25})$. On the other hand, Liu et al. [34] employed a single bcc-A2 phase to describe the bcc-phase region using the thermodynamic parameters reported by Shim et al. [58], in which the $D0_3$ phase was not considered, already accepted in the COST 531 (European Cooperation in the Field of Scientific and Technical research (COST) Program) lead-free solders database [4]. Although these calculated results [21,58–64] are in agreement with the reported experimental results, the bcc-A2 and $D0_3$ phases as well as the intermetallic compounds with composition ranges are not main concerns for Sn-Ag-Cu solders and Ag-Cu-Sn-based brazing alloys, and quite complex models are used in these thermodynamic calculations of the Cu-Sn binary system, which is inconvenient to extrapolate and develop a thermodynamic database of the multi-component Ag-Cu-Sn-based alloy systems. Therefore, the calculated results by Liu et al. [34] were used in this work. The calculated Cu-Sn binary phase diagram is shown in Figure 1c.

## 2.4. The Ag-Cu-Sn Ternary System

Phase equilibria of the Ag-Cu-Sn ternary system including liquidus projection, isothermal sections at different temperatures and vertical sections of different compositions were studied experimentally [18–27]. The experimental results [18–27] summarized in Table 1 do not include the stable ternary intermetallic compounds. According to the thermodynamic calculations of the Ag-Cu-Sn ternary system [21,24,25,28], thermodynamic properties of the ternary liquid alloys were not considered in the calculated results of Moon et al. [21], Ohnuma et al. [24] and Yen et al. [25], while the isothermal sections and vertical sections of phase relations calculated by Gierlotka et al. [28] were obviously different from the determined experimental results [18,24–27].

**Table 1.** Experimental information in the Ag-Cu-Sn ternary system.

| Reference | Experimental Method * | Experimental Data | Used in Optimization |
|---|---|---|---|
| [18] | DTA, XRD | Liquidus projection<br>Vertical sections: 5 wt%, 10 wt%, 15 wt%, 20 wt%, 25 wt% Sn<br>Isothermal sections: 773 K, 873 K | Yes<br>Yes<br>Yes |
| [19] | DTA, XRD, OM, SEM | Ternary eutectic reaction | Yes |
| [20] | DSC, OM, SEM | Ternary eutectic reaction | Yes |
| [21] | DTA, SEM | Ternary eutectic reaction | Yes |
| [22] | SEM, EPMA | Ternary eutectic reaction | Yes |
| [23] | DSC, SEM | Ternary eutectic reaction | Yes |
| [24] | XRD, SEM | Isothermal sections: 573 K, 673 K, 773 K | Yes |
| [25] | XRD, OM, SEM, EPMA | Isothermal sections: 513 K, 723 K | Yes |
| [26] | DTA | Vertical sections:<br>$x_{Ag} : x_{Cu} = 1$, $x_{Ag} : x_{Sn} = 1$, $x_{Cu} : x_{Sn} = 1$ | Yes |
| [27] | DSC | Vertical sections:<br>$x_{Ag} : x_{Cu} = 1$, $x_{Ag} : x_{Cu} = 7 : 3$ | Yes |
| [30] | Calorimetry | Enthalpy of mixing of liquid at 773 K, 973 K, 1173 K for three constant ratios of $x_{Ag} : x_{Cu} = 1 : 3$, $x_{Ag} : x_{Cu} = 3 : 1$, $x_{Ag} : x_{Cu} = 1$ | Yes |
| [31] | Electromotive force (EMF) | Activity of Sn in liquid 1000 K and 1300 K for three constant ratios of $x_{Ag} : x_{Cu} = 1 : 3$, $x_{Ag} : x_{Cu} = 3 : 1$, $x_{Ag} : x_{Cu} = 1$ | Yes |

* DTA = Differential thermal analysis, DSC = differential scanning calorimeter, XRD = X-Ray Diffraction, OM = Optical metallography, SEM = Scanning electron microscopy, EPMA = Electron probe microanalysis.

The liquidus projection of the Ag-Cu-Sn ternary system in the Ag-Cu-rich part was studied experimentally by Gebhardt et al. [18]. The eutectic reaction (L $\leftrightarrow$ Ag$_3$Sn + Cu$_6$Sn$_5$-h + bct(Sn)) in the Ag-Cu-Sn ternary system was studied experimentally [19–23]. Miller et al. [19] measured the composition and temperature of this ternary eutectic reaction to be Sn-4.7 wt.% Ag-1.7 wt.% Cu and 490.0 K, using optical microscopy (OM), scanning electron microscopy (SEM) and X-ray powder diffraction (XRD). Loomans et al. [20] determined the composition and temperature of this ternary eutectic reaction to be Sn-3.5 wt.% Ag-0.9 wt.% Cu and 490.3 K, which is in agreement with the results measured by Miller et al. [18], Moon et al. [21], Lewis et al. [22] and Park et al. [23]. Two isothermal sections at 773 K and 873 K as well as five vertical sections at 5 wt.% Sn, 10 wt.% Sn, 15 wt.% Sn, 20 wt.% Sn and 25 wt.% Sn were determined by Gebhardt et al. [18] using metallographic microscopy, XRD and differential thermal analysis (DTA). Ohnuma et al. [24] measured three isothermal sections at 573 K, 673 K and 773 K using metallography, SEM and XRD, while Yen et al. [25] determined two isothermal sections at 513 K and 723 K using electronic probe microanalysis (EPMA), OM, XRD and SEM. Marjanovic et al. [26] determined three vertical sections of $x_{Ag} : x_{Cu} = 1$, $x_{Ag} : x_{Sn} = 1$ and $x_{Cu} : x_{Sn} = 1$ by thermal analysis, and Fima et al. [27] measured two vertical sections of $x_{Ag} : x_{Cu} = 1$ and $x_{Ag} : x_{Cu} = 7 : 3$ using thermal analysis.

Thermodynamic properties of the Ag-Cu-Sn ternary system were investigated experimentally [30,31]. Luef et al. [30] measured the enthalpy of mixing of liquid Ag-Cu-Sn alloys with different composition ratios ($x_{Ag} : x_{Cu} = 1 : 3$, $x_{Ag} : x_{Cu} = 3 : 1$, $x_{Ag} : x_{Cu} = 1$) at 773 K, 973 K and 1173 K using a Calvet-type microcalorimeter. The activity of Sn ($x_{Ag} : x_{Cu} = 1 : 3$, $x_{Ag} : x_{Cu} = 3 : 1$, $x_{Ag} : x_{Cu} = 1$) in liquid Ag-Cu-Sn alloys at 1000 K and 1300 K was determined by Kopyto et al. [31] by the electromotive force method.

## 3. Thermodynamic Models

### 3.1. Solution Phases

The solution phase $\Phi$ including Liquid, fcc, bcc, hcp, bct(Sn) and diamond(Sn) is described using the substitutional solution model. The molar Gibbs energy of the solution phase $\Phi$ can be expressed as:

$$G_m^\Phi = \sum_{i=Ag,Cu,Sn} x_i^0 G_i^\Phi + RT \sum_{i=Ag,Cu,Sn} x_i ln x_i + {}^{ex}G_m^\Phi \tag{1}$$

where ${}^0 G_i^\Phi$ is the molar Gibbs energy of the element $i$ ($i$ = Ag, Cu, Sn) with the structure $\Phi$ referring to the enthalpy of its stable state at 298.15 K and 1 bar, and ${}^{ex}G_m^\Phi$ is the excess Gibbs energy, $x_i$ is the mole fraction of the element $i$, and $R$ is the gas constant and $T$ is temperature. In this work, ${}^0 G_i^\Phi(T)$ is taken from the Science Group Thermodata Europe (SGTE) compiled by Dinsdale [56]. It is noted that the phase stabilities of hcp-Sn phase used in this work is from the new version of the pure elements database [21]. The excess Gibbs energy of the solution phase $\Phi$ can be expressed by the Redlich–Kister–Muggianu polynomial [65,66] as:

$$\begin{aligned} {}^{ex}G_m^\Phi = x_{Ag}x_{Cu}\sum_{j=0}^{n} {}^j L_{Ag,Cu}^\Phi (x_{Ag}-x_{Cu})^j + x_{Ag}x_{Sn}\sum_{j=0}^{n} {}^j L_{Ag,Sn}^\Phi (x_{Ag}-x_{Sn})^j + x_{Cu}x_{Sn}\sum_{j=0}^{n} {}^j L_{Cu,Sn}^\Phi (x_{Cu}-x_{Sn})^j \\ + x_{Ag}x_{Cu}x_{Sn}(x_{Ag}{}^0 L_{Ag,Cu,Sn}^\Phi + x_{Cu}{}^1 L_{Ag,Cu,Sn}^\Phi + x_{Sn}{}^2 L_{Ag,Cu,Sn}^\Phi) \end{aligned} \tag{2}$$

where ${}^j L_{Ag,Cu}^\Phi$, ${}^j L_{Ag,Sn}^\Phi$ and ${}^j L_{Cu,Sn}^\Phi$ are the interaction parameters and are taken from the Ag-Cu, Ag-Sn and Cu-Sn binary systems assessed by He et al. [32], Du et al. [33] and Liu et al. [34], respectively. The parameters ${}^0 L_{Ag,Cu,Sn}^\Phi$, ${}^1 L_{Ag,Cu,Sn}^\Phi$ and ${}^2 L_{Ag,Cu,Sn}^\Phi$ represent the ternary interaction parameters to be assessed in this work.

### 3.2. Intermetallic Compounds

Based on the experimental results determined [18,24,25], the solubility of the third element in the intermetallic compounds in the Cu-Sn and Ag-Sn binary systems is negligible. Therefore, the intermetallic compounds $Cu_3Sn$, $Cu_{10}Sn_3$, $Cu_{41}Sn_{11}$, $Cu_6Sn_5$-h and $Cu_6Sn_5$-l were treated as the stoichiometric compounds. The molar Gibbs energies of these intermetallic compounds were expressed as:

$$G_m^{Cu_3Sn} = 0.75^0 G_{Cu}^{fcc} + 0.25^0 G_{Sn}^{bct} + A_1 + B_1 T \tag{3}$$

$$G_m^{Cu_{10}Sn_3} = 0.769^0 G_{Cu}^{fcc} + 0.231^0 G_{Sn}^{bct} + A_2 + B_2 T \tag{4}$$

$$G_m^{Cu_{41}Sn_{11}} = 0.788^0 G_{Cu}^{fcc} + 0.212^0 G_{Sn}^{bct} + A_3 + B_3 T \tag{5}$$

$$G_m^{Cu_6Sn_5-h} = 0.545^0 G_{Cu}^{fcc} + 0.455^0 G_{Sn}^{bct} + A_4 + B_4 T \tag{6}$$

$$G_m^{Cu_6Sn_5-l} = 0.545^0 G_{Cu}^{fcc} + 0.455^0 G_{Sn}^{bct} + A_5 + B_5 T \tag{7}$$

where ${}^0 G_{Ag}^{fcc}$, ${}^0 G_{Cu}^{fcc}$ and ${}^0 G_{Sn}^{bct}$ are the Gibbs energies of the pure elements Ag, Cu and Sn. The parameters $A_1$, $A_2$, $A_3$, $A_4$, $A_5$ and $B_1$, $B_2$, $B_3$, $B_4$, $B_5$ were taken from the assessed results of the Cu-Sn binary systems [34].

The intermetallic compound $Ag_3Sn$ was modeled by the two-sublattice model $(Ag)_{0.75}(Ag,Sn)_{0.25}$ to maintain the consistency with other databases by unifying the models for phases with the same crystallographic structure. The molar Gibbs energy of $Ag_3Sn$ was expressed as:

$$\begin{aligned} G_m^{Ag_3Sn} = Y_{Ag}^I Y_{Ag}^{II} G_{Ag:Ag}^{Ag_3Sn} + Y_{Ag}^I Y_{Sn}^{II} G_{Ag:Sn}^{Ag_3Sn} + 0.75 RT Y_{Ag}^I \ln Y_{Ag}^I + 0.25 RT \left( Y_{Ag}^{II} \ln Y_{Ag}^{II} + Y_{Sn}^{II} \ln Y_{Sn}^{II} \right) \\ + Y_{Ag}^I Y_{Ag}^{II} Y_{Sn}^{II} L_{Ag:Ag,Sn}^{Ag_3Sn} \end{aligned} \tag{8}$$

where $Y_i^I$ and $Y_i^{II}$ denote mole fraction of $i$ ($i = Ag, Sn$) in the first and second sublattice, respectively. The parameters $G_{Ag:Ag}^{Ag_3Sn}$, $G_{Ag:Sn}^{Ag_3Sn}$ and $L_{Ag:Ag,Sn}^{Ag_3Sn}$ are directly taken from the Ag-Sn binary system [33].

## 4. Results and Discussion

Based on the reported experimental information including phase equilibria data and thermodynamic data, thermodynamic parameters of the various phases (especially for liquid phase) in the Ag-Cu-Sn ternary system were optimized using the PARROT module in the Thermo-Calc® software (Thermo-Calc Software AB, Solna, Sweden) package developed by Sundman et al. [67]. The final thermodynamic parameters of all the phases in the Ag-Cu-Sn ternary system are shown in Table 2.

**Table 2.** Thermodynamic parameters for the Ag-Cu-Sn ternary system.

| Phase | Thermodynamic Parameters | Reference |
|---|---|---|
| Liquid | $^0L_{Ag,Cu}^{liq} = 16914.949 - 14.7721T + 1.54955TlnT$ | [32] |
| | $^1L_{Ag,Cu}^{liq} = -1963.300 + 0.8623T$ | [32] |
| | $^0L_{Ag,Sn}^{liq} = -3177.49 - 10.16124T + 0.380505TlnT$ | [33] |
| | $^1L_{Ag,Sn}^{liq} = -16782.28 + 2.06521T + 0.437477TlnT$ | [33] |
| | $^2L_{Ag,Sn}^{liq} = 3190.34 - 107.09456T + 13.954838TlnT$ | [33] |
| | $^0L_{Cu,Sn}^{liq} = -9002.8 - 5.8381T$ | [34] |
| | $^1L_{Cu,Sn}^{liq} = -20100.4 + 3.63666T$ | [34] |
| | $^2L_{Cu,Sn}^{liq} = -10528.40$ | [34] |
| | $^0L_{Ag,Cu,Sn}^{liq} = -80000 + 27.9828T$ | This work |
| | $^1L_{Ag,Cu,Sn}^{liq} = -85233 + 29.4392T$ | This work |
| | $^2L_{Ag,Cu,Sn}^{liq} = -40000 + 26.5606T$ | This work |
| fcc | $^0L_{Ag,Cu}^{fcc} = 32580.365 - 7.4547T$ | [32] |
| | $^1L_{Ag,Cu}^{fcc} = -10144.596 + 5.5617T$ | [32] |
| | $^0L_{Ag,Sn}^{fcc} = 745.45 + 11.498027T$ | [33] |
| | $^1L_{Ag,Sn}^{fcc} = -36541.5$ | [33] |
| | $^0L_{Cu,Sn}^{fcc} = -10672 - 1.4837T$ | [34] |
| | $^1L_{Cu,Sn}^{fcc} = -15331.3 + 6.9539T$ | [34] |
| bct | $^0L_{Ag,Sn}^{bct} = 18358.8$ | [33] |
| hcp | $^0L_{Ag,Sn}^{hcp} = 1046.1 + 10.23693T$ | [33] |
| | $^1L_{Ag,Sn}^{hcp} = -40505.5$ | [33] |
| bcc | $^0L_{Cu,Sn}^{bcc} = -32656.8 + 25.0158T$ | [34] |
| | $^1L_{Cu,Sn}^{bcc} = -13862.5 - 32.0218T$ | [34] |
| | $^2L_{Cu,Sn}^{bcc} = -4175.47 + 5.0083T$ | [34] |
| | $^0L_{Ag,Cu}^{bcc} = 10100$ | This work |
| Ag₃Sn | $^0G_{Ag,Ag}^{Ag3Sn} = 4750 - 0.5T + {}^0G_{Ag}^{fcc}$ | [33] |
| | $^0G_{Ag,Sn}^{Ag3Sn} = -11085.3 + 110.01471T - 23.18TlnT - 0.00359T^{-2} + 4389.5T^{-1}$ | [33] |
| Cu₃Sn | $^0G_{Cu,Sn}^{Cu3Sn} = -8194.2 - 0.2043T + 0.75{}^0G_{Cu}^{fcc} + 0.25{}^0G_{Sn}^{bct}$ | [34] |
| Cu₁₀Sn₃ | $^0G_{Cu,Sn}^{Cu10Sn3} = -6655.1 - 1.485T + 0.769{}^0G_{Cu}^{fcc} + 0.231{}^0G_{Sn}^{bct}$ | [34] |
| Cu₄₁Sn₁₁ | $^0G_{Cu,Sn}^{Cu41Sn11} = -6323.5 - 1.2808T + 0.788{}^0G_{Cu}^{fcc} + 0.212{}^0G_{Sn}^{bct}$ | [34] |
| Cu₆Sn₅-h | $^0G_{Cu,Sn}^{Cu6Sn5-h} = -6869.5 - 0.1589T + 0.545{}^0G_{Cu}^{fcc} + 0.455{}^0G_{Sn}^{bct}$ | [34] |
| Cu₆Sn₅-l | $^0G_{Cu,Sn}^{Cu6Sn5-l} = -7129.7 + 0.4059T + 0.545{}^0G_{Cu}^{fcc} + 0.455{}^0G_{Sn}^{bct}$ | [34] |

Figure 2 is the calculated liquidus projection of the Ag-Cu-Sn ternary system with the calculated results [21,24,28]. The calculated reaction scheme related to the liquid phase

in the Ag-Cu-Sn ternary system is given in Figure 3. Figures 2 and 3 show five U-type reactions ($U_1$: L+ fcc(Cu) ↔ fcc(Ag) + bcc at 891.5 K, $U_2$: L+ bcc ↔ fcc(Ag) + $Cu_3Sn$ at 849.2 K, $U_3$: L+ fcc(Ag) ↔ hcp + $Cu_3Sn$ at 810.0 K, $U_4$: L+ hcp ↔ $Ag_3Sn$ + $Cu_3Sn$ at 719.6 K, $U_5$: L+ $Cu_3Sn$ ↔ $Ag_3Sn$ + $Cu_6Sn_5$-h at 623.8 K) and one E-type reaction (E: L ↔ $Ag_3Sn$ + $Cu_6Sn_5$-h + bct(Sn) at 490.3 K). The calculated results by Moon et al. [21] and Gierlotka et al. [28] did not have the $U_2$ and $U_3$ reactions because the different thermodynamic models of the bcc-A2 and $D0_3$ phases in the Cu-Sn binary system were used in their calculations. The comparison of the calculated invariant reactions in this work with the experimental data [18–23] and the calculated results [21,24,28] is shown in Table 3. The calculated compositions and temperatures of the eutectic reaction (L ↔ $Ag_3Sn$ + $Cu_6Sn_5$-h + bct(Sn)) are in good agreement with the experimental results [18–23]. The calculated liquidus projection in this work is similar to those the results determined by Ohnuma et al. [24].

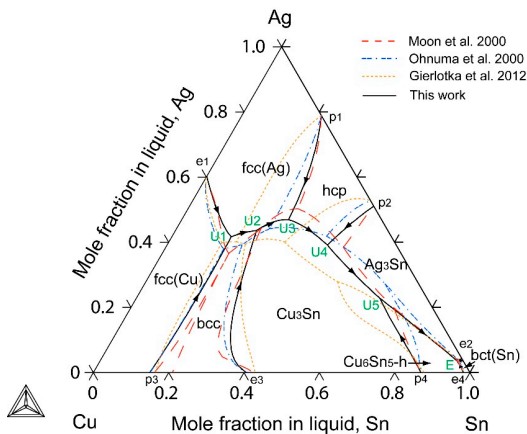

**Figure 2.** The calculated liquidus projection of the Ag-Cu-Sn ternary system with the calculations from [21,24,28].

Figure 4 shows the calculated isothermal sections at different temperatures (513 K, 573 K, 673 K, 723 K, 773 K and 873 K) with the experimental results [18,24,25] and the calculated results [21,24,25]. Apart from solid solubility, the calculated phase relations of the isothermal section at 513 K in Figure 4a are consistent with the experimental results [25], while the calculated boundaries of the three-phase region ($Ag_3Sn$ + $Cu_3Sn$ + hcp) show a slight deviation from the experimental results [25]. As shown in Figures 4b and 4c, the calculated phase relations of isothermal sections at 573 K and 673 K in this work can reproduce well with the experimental results determined by Ohnuma et al. [24], without considering the solid solubility. However, the calculated isothermal section at 773 K by Ohnuma et al. [24] has the bcc phase in Figure 4e, indicating that it is unreliable because the bcc phase does not appear at 773 K according to the Cu-Sn binary phase diagram [63]. In Figure 4d, apart from solid solubility, the calculated isothermal section at 723 K in this work is in good agreement with the experimental results measured by Yen et al. [25], and the liquid-phase region calculated by Yen et al. [25] at 723 K is quite different from the calculations [21,24] because the thermodynamic parameters of the liquid phase were not considered. In Figure 4f, the calculated isothermal section at 873 K in this work agrees well with the experimental results [18]. The calculated results in this work are in much better agreement with the experimental results [18,24,25] than the previous calculations [21,24,25].

Figure 5 is the calculated vertical sections with different molar ratios ($x_{Ag} : x_{Cu} = 1$, $x_{Ag} : x_{Sn} = 1$, $x_{Cu} : x_{Sn} = 1$, $x_{Ag} : x_{Cu} = 7 : 3$) with the experimental results [26,27] and the calculated results [21,24]. The calculated vertical section of $x_{Ag} : x_{Cu} = 1$ in Figure 5a is in good agreement with the experimental results determined by Marjanovic et al. [26] and Fima et al. [27], while the calculated vertical sections of $x_{Ag} : x_{Sn} = 1$ and $x_{Cu} : x_{Sn} = 1$ in Figure 5b and c are consistent with the experimental results [26]. The calculated liquidus by Moon et al. [21] is slightly higher than the experimental data [26], while the

calculated liquidus by Ohnuma et al. [23] is slightly lower than the experimental data [26]. In Figure 5d, the calculated vertical section of $x_{Ag} : x_{Cu} = 7 : 3$ in this work agrees well with the experimental results [27], while the calculated liquidus by Moon et al. [21] and Ohnuma et al. [24] are slightly higher than the experimental data [27]. The calculated results in this work are in much better agreement with the experimental results [26,27] than the previous calculations [21,24].

Figure 6 shows the calculated five vertical sections with different Sn contents (Sn = 5, 10, 15, 20, 25 wt.%) with the experimental data [18] and the calculated results [21,24]. The calculated vertical sections of this work agreed well with the experimental results measured by Gebhardt et al. [18]. The calculated results in this work are similar to the calculated results [21,24] and are in better agreement with the experimental data [18,26,27].

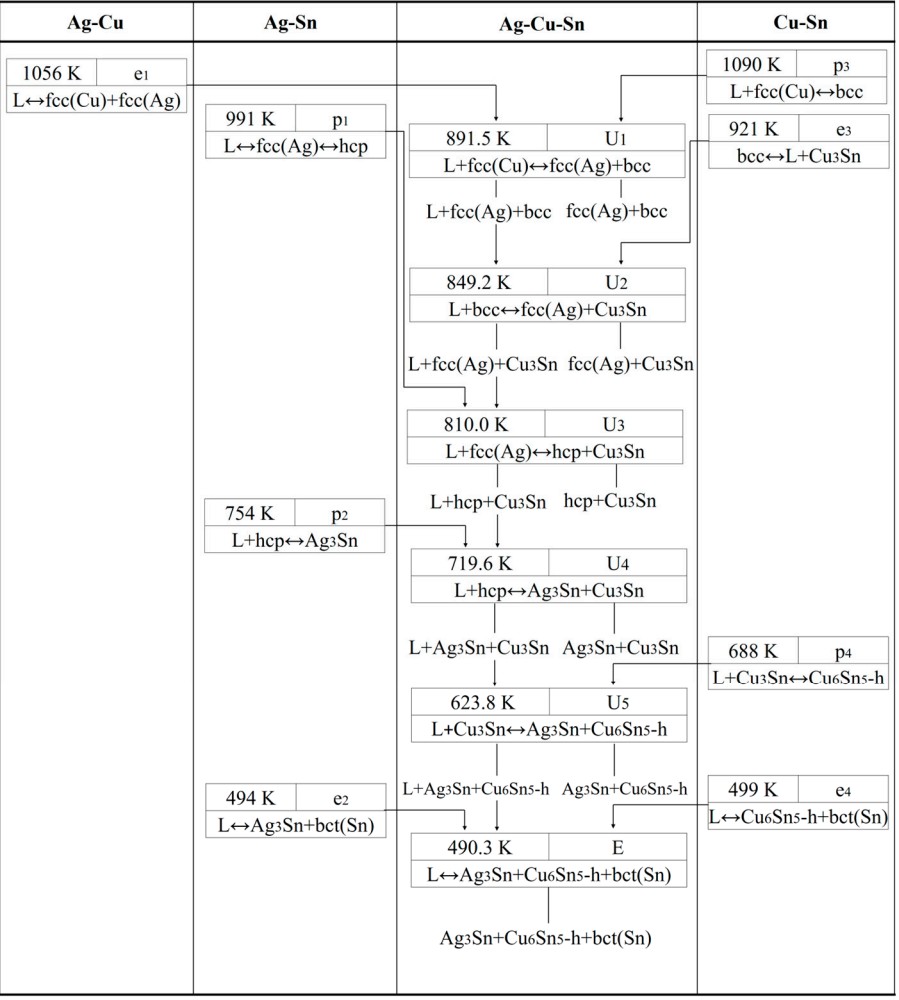

**Figure 3.** The reaction scheme in the Ag-Cu-Sn ternary system.

Figure 7 is the calculated enthalpies of mixing of the liquid Ag-Cu-Sn alloys at 773 K, 973 K and 1173 K (referenced states: liquid Ag, liquid Cu and liquid Sn) along different cross sections ($x_{Ag} : x_{Cu} = 1 : 3$, $x_{Ag} : x_{Cu} = 3 : 1$, $x_{Ag} : x_{Cu} = 1 : 1$) with the experimental data measured by Luef et al. [30] and the calculated results [21,24,28]. The calculated results (solid line) at 1173 K in this work fit well with the experimental data compared with the previously calculated results (dashed lines) [21,24,28]. The calculated results by Gierlotka et al. [28] have a good fit with the experimental data at 973 K, and show a slight deviation from the experimental data at 773 K. Nonetheless, good agreements are achieved between the calculated enthalpies of mixing of the liquid Ag-Cu-Sn alloys

and the experimental data at 1173 K in this work, which are better than the calculated results [21,24,28].

**Table 3.** Invariant reactions in the Ag-Cu-Sn ternary system.

| Invariant Reactions | Type | T(K) | Composition | | Reference |
|---|---|---|---|---|---|
| | | | $x^L_{Ag}$ | $x^L_{Cu}$ | |
| L + fcc(Cu) ↔ bcc + fcc(Ag) | $U_1$ | 891.5 | 0.417 | 0.425 | This work |
| | | 878 | 0.545 | 0.224 | Exp. [18] |
| | | 910 | 0.366 | 0.457 | Cal. [21] |
| | | 841 | 0.380 | 0.464 | Cal. [24] |
| | | 857 | 0.401 | 0.460 | Cal. [28] |
| L + bcc ↔ fcc(Ag) + Cu$_3$Sn | $U_2$ | 849.2 | 0.436 | 0.345 | This work |
| | | 833 | 0.492 | 0.192 | Exp. [18] |
| | | 828 | 0.392 | 0.409 | Cal. [24] |
| L + fcc(Ag) ↔ hcp + Cu$_3$Sn | $U_3$ | 810.0 | 0.469 | 0.247 | This work |
| | | 813 | 0.494 | 0.152 | Exp. [18] |
| | | 845 | 0.503 | 0.206 | Cal. [21] |
| | | 814 | 0.441 | 0.313 | Cal. [24] |
| L + hcp ↔ Ag$_3$Sn + Cu$_3$Sn | $U_4$ | 719.6 | 0.391 | 0.183 | This work |
| | | 713 | 0.437 | 0.071 | Exp. [18] |
| | | 733 | 0.385 | 0.144 | Cal. [21] |
| | | 721 | 0.419 | 0.172 | Cal. [24] |
| | | 741 | 0.400 | 0.292 | Cal. [28] |
| L + Cu$_3$Sn ↔ Ag$_3$Sn + Cu$_6$Sn$_5$-h | $U_5$ | 623.8 | 0.227 | 0.125 | This work |
| | | 623 | 0.191 | 0.034 | Exp. [18] |
| | | 629 | 0.198 | 0.102 | Cal. [21] |
| | | 645 | 0.272 | 0.089 | Cal. [24] |
| | | 632 | 0.281 | 0.215 | Cal. [28] |
| L ↔ Ag$_3$Sn + Cu$_6$Sn$_5$-h + bct(Sn) | E | 490.3 | 0.035 | 0.016 | This work |
| | | 498.2 | 0.036 | 0.003 | Exp. [18] |
| | | 490.0 | 0.051 | 0.031 | Exp. [19] |
| | | 490.3 | 0.038 | 0.016 | Exp. [20] |
| | | 490.3 | 0.038 | 0.016 | Exp. [21] |
| | | 490.3 | 0.038 | 0.016 | Exp. [22] |
| | | 490.2 | 0.039 | 0.016 | Exp. [23] |
| | | 489.5 | 0.041 | 0.017 | Cal. [21] |
| | | 490.7 | 0.035 | 0.011 | Cal. [24] |
| | | 491.9 | 0.039 | 0.003 | Cal. [28] |

Figure 8 is the calculated activity of Sn in liquid Ag-Cu-Sn alloys at 1000 K and 1300 K with the experimental data determined by Kopyto et al. [31] and the calculated results [21,24,28]. It was shown that the calculated results in this work (solid line) are reasonably consistent with the experimental data [31] and are better than the calculated results [21,24,28].

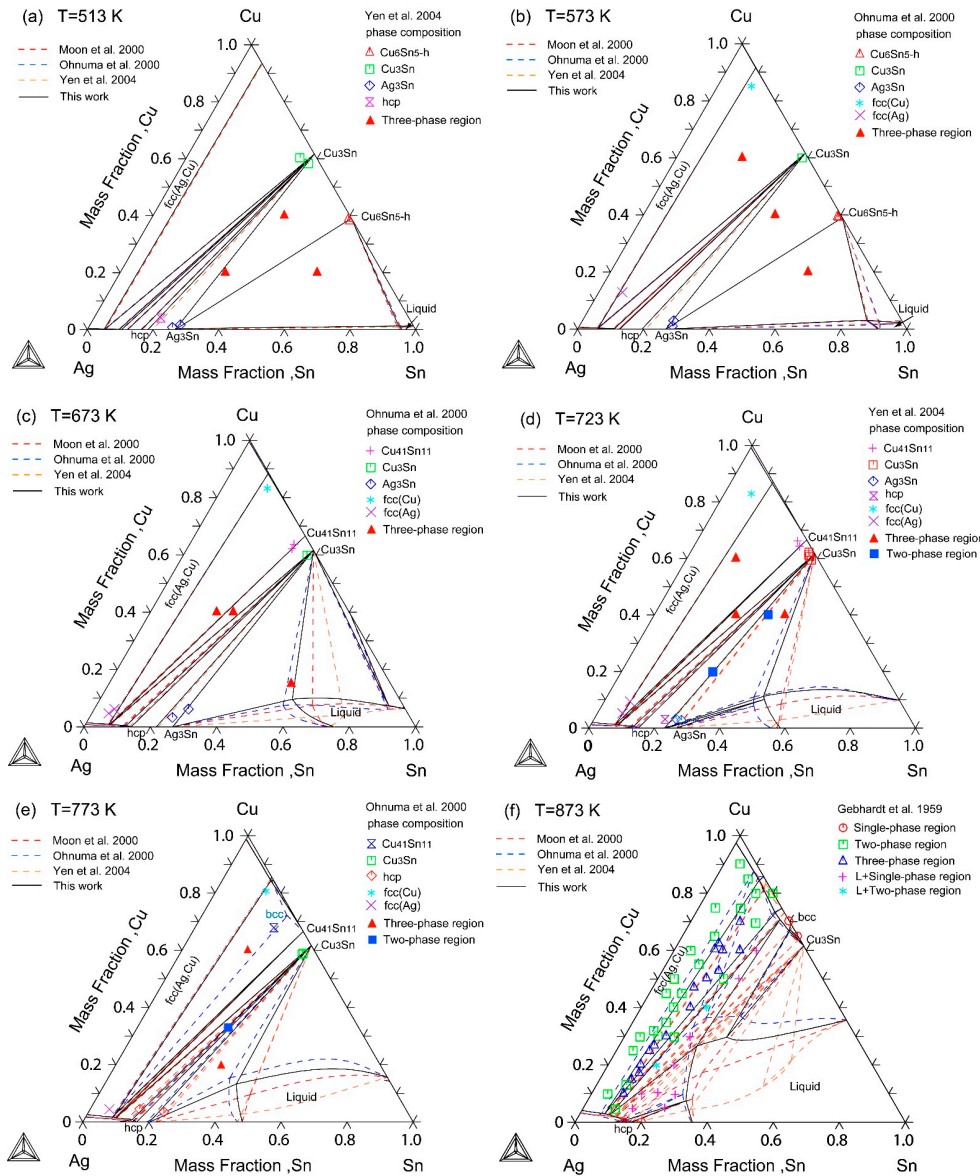

**Figure 4.** Calculated isothermal sections of the Ag-Cu-Sn ternary system at different temperatures with the experimental data from [18,24,25] and the calculated results from [21,24,25]. (**a**) 513 K; (**b**) 573 K; (**c**) 673 K; (**d**) 723 K; (**e**) 773 K; (**f**) 873 K.

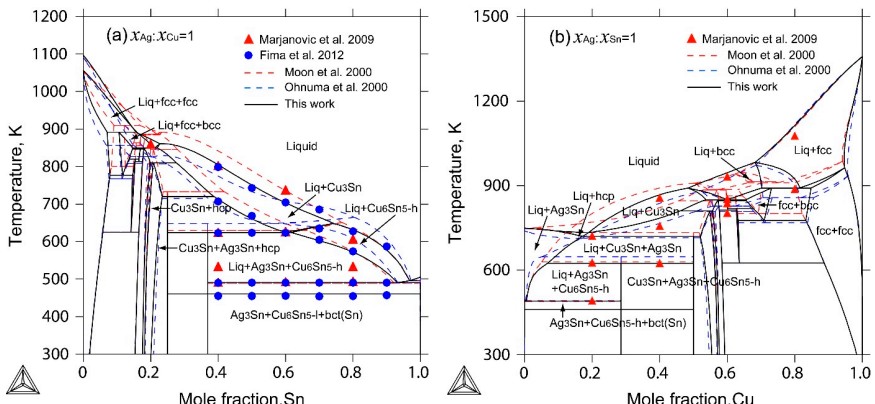

**Figure 5.** *Cont.*

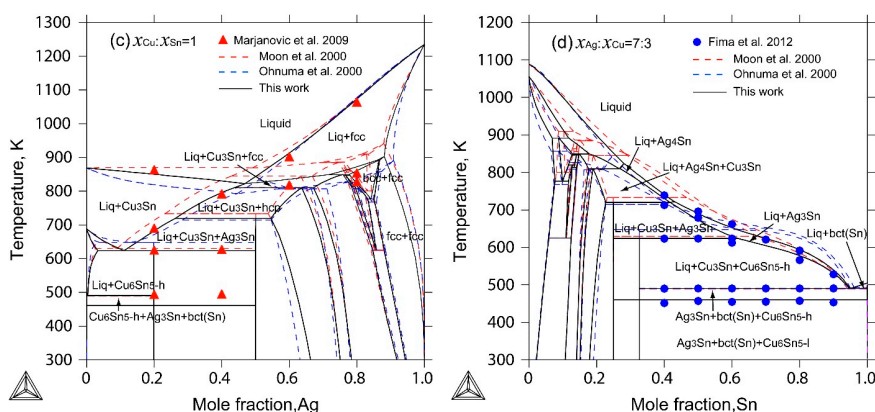

**Figure 5.** Calculated vertical sections of the Ag-Cu-Sn ternary system with the experimental data from [26,27] and the calculated results from [21,24]. (**a**) $x_{Ag} : x_{Cu} = 1$; (**b**) $x_{Ag} : x_{Sn} = 1$; (**c**) $x_{Cu} : x_{Sn} = 1$; (**d**) $x_{Ag} : x_{Cu} = 7 : 3$.

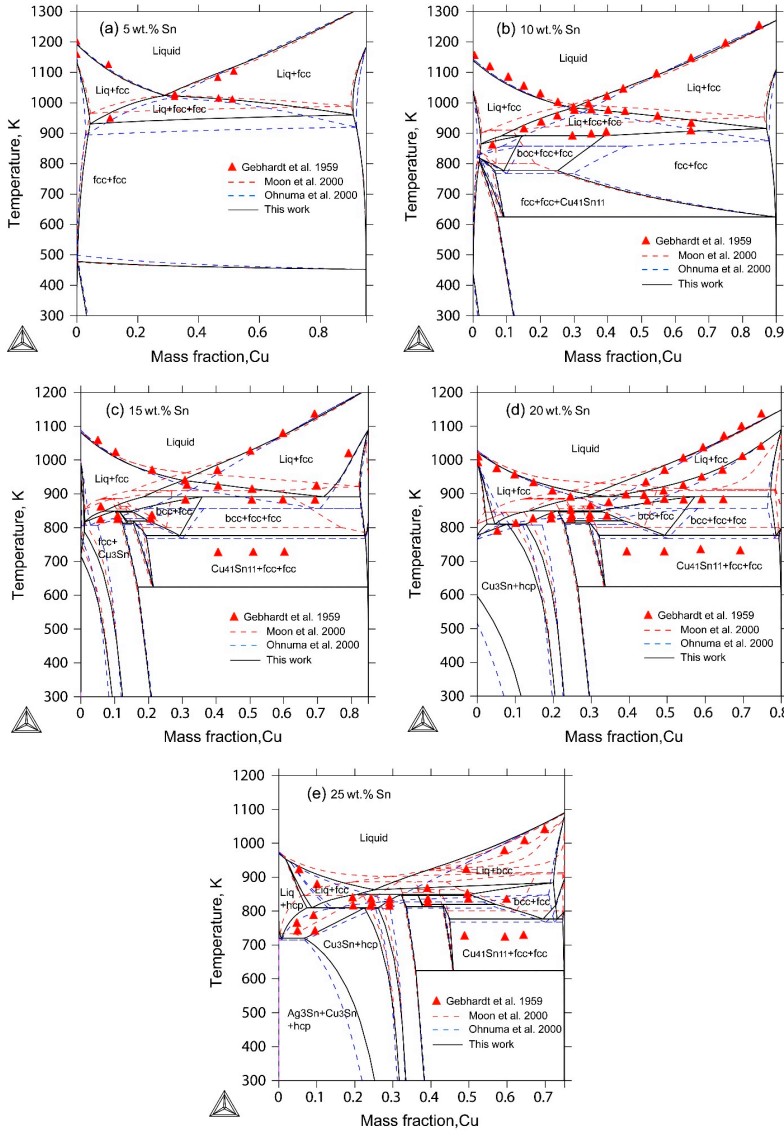

**Figure 6.** Calculated vertical sections of the Ag-Cu-Sn ternary system with the experimental data from [18] and the calculated results from [21,24]. (**a**) $w_{Sn} = 5$ %; (**b**) $w_{Sn} = 10$ %; (**c**) $w_{Sn} = 15$ %; (**d**) $w_{Sn} = 20$ %; (**e**) $w_{Sn} = 25$ %.

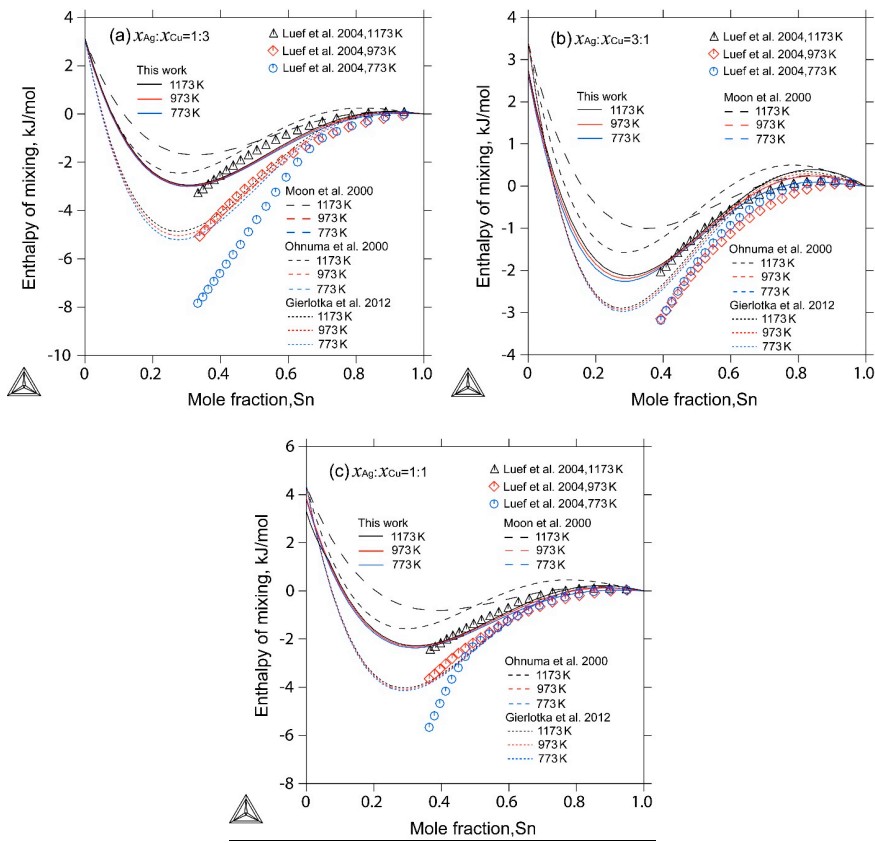

**Figure 7.** Comparison of the calculated enthalpies of mixing of Ag-Cu-Sn liquid alloys with the experimental data from [30] and the calculated results from [21,24,28] at 773 K, 973 K and 1173 K (referenced states: liquid Ag, liquid Cu and liquid Sn). (**a**) $x_{Ag} : x_{Cu} = 1 : 3$; (**b**) $x_{Ag} : x_{Cu} = 3 : 1$; (**c**) $x_{Ag} : x_{Cu} = 1$.

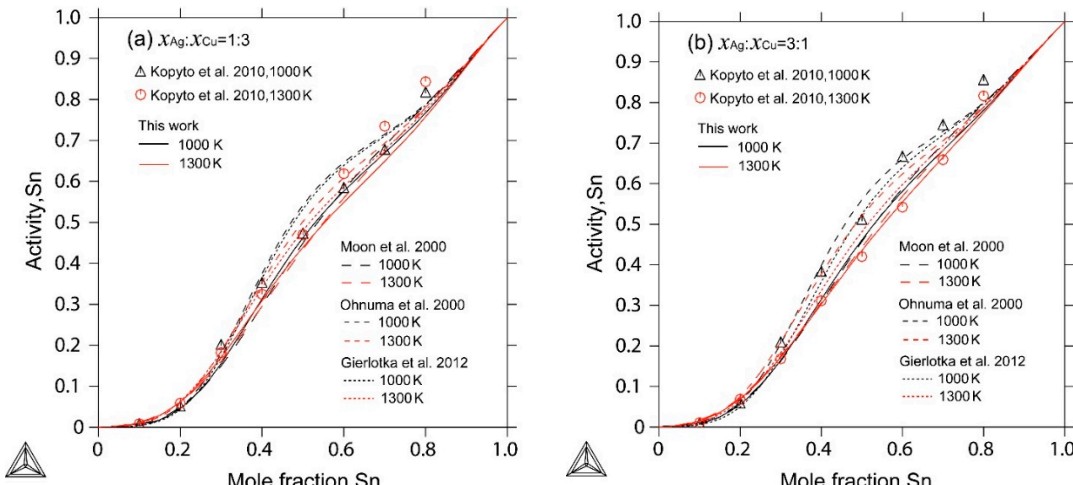

**Figure 8.** *Cont.*

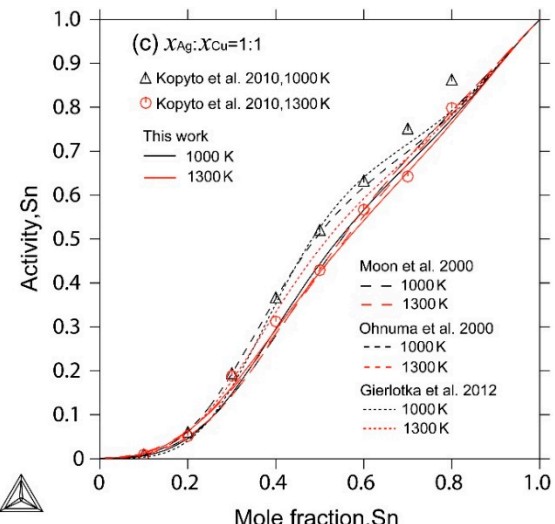

**Figure 8.** Comparison of the calculated activities of Sn in liquid ternary alloys with the experimental data from [31] and the calculated results from [21,24,28] at 1000 K and 1300 K (referenced states: liquid Ag, liquid Cu and liquid Sn). (**a**) $x_{Ag} : x_{Cu} = 1 : 3$; (**b**) $x_{Ag} : x_{Cu} = 3 : 1$; (**c**) $x_{Ag} : x_{Cu} = 1$.

## 5. Conclusions

On the basis of the previous assessments of the Ag-Cu, Ag-Sn and Cu-Sn binary systems, thermodynamic modeling of the Ag-Cu-Sn ternary system was performed using the CALPHAD method with the reported experimental results of phase diagram data and thermodynamic data. The results are summarized as follows:

- A set of self-consistent thermodynamic parameters formulating the Gibbs energies of various phases in the Ag-Cu-Sn ternary system were obtained in this work.
- The liquidus projection, vertical sections and isothermal sections were calculated, which are in good agreement with the reported experimental results, assuming that all binary intermetallic compounds have no ternary solubility.
- Thermodynamic parameters of the Ag-Cu-Sn ternary system obtained in this work provide a good basis to develop a compatible thermodynamic database including multicomponent Ag-Cu-Sn-based alloys.

**Author Contributions:** Q.T., M.R. and J.W. conceived and designed the calculations; Q.T. and J.G. performed the calculations; J.L., J.J. and L.Z. analyzed the data; Q.T., M.R. and J.W. wrote the paper. All authors have read and agreed to the published version of the manuscript.

**Funding:** This research was funded by China (151009-Z) and Guangxi Undergraduate Training Program for Innovation (S202110595243).

**Data Availability Statement:** All the data that support the findings of this study are included within the article.

**Acknowledgments:** This work was supported financially by Guangxi Key Laboratory of Information Materials, Guilin University of Electronic Technology, China (151009-Z) and Guangxi Undergraduate Training Program for Innovation (S202110595243).

**Conflicts of Interest:** The authors declare no conflict of interest.

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
