# Peer review of "Thermodynamic Modeling of the Ag-Cu-Sn Ternary System"

_metals, doi:10.3390/met12101557_

Round 1

Reviewer 1 Report

The paper describes thermodynamic modelling of the Ag-Cu-Sn system using Thermo-Calc. The normal methods were used, but in some places it was not clear whether certain published data were used or not. I could not find any reference to the assessment in COST 531, which should be included.

It was stated more than once that only the ternary interaction parameters were assessed in this work, and that no modelling was done on the extensions of the solid phases because no data were available. However, data were plotted from other researchers in Figure 4.

The agreement with prior work was not always as goos as the authors stated.

For ease of clarity, the reference numbers must also be given in the legends of all the diagrams when prior work is plotted for comparision (Figures 2, 4, 5, 6, 7 and 8). This will make the paper much easier to follow. 

It should be stated in the conclusions that the solid solubilities were not taken into account, although they should be.

More should be stated about the bcc-A2 phase in Cu-Sn. It was stated that a ternary was apparently incorrect, but no further reason was given for stating that the binary was correct.

The temperatures of some of the curves should be given on Figure 8.

The writing style is long-winded, and the paper should be written in a more succinct way. There are also places where more clarity is needed. These and other corrections are given on the attacted scan.

Author Response

Response to the reviewers

The authors thank the reviewers for the helpful comments and suggestions. In the following, these comments and questions are addressed in detail.

Reviewer #1

Q1. The paper describes thermodynamic modelling of the Ag-Cu-Sn system using Thermo-Calc. The normal methods were used, but in some places it was not clear whether certain published data were used or not. I could not find any reference to the assessment in COST 531, which should be included.

Answer: Thanks for the suggestion. The related references for the assessment in COST 531  were cited in the text (e.g. Refs. [4,29]). Please check them in the revised manuscript.

Q2. It was stated more than once that only the ternary interaction parameters were assessed in this work, and that no modelling was done on the extensions of the solid phases because no data were available. However, data were plotted from other researchers in Figure 4.

Answer: Thanks for the comment. The solubility of the third element in binary intermetallic compounds was not taken into account because the solubility for many binary intermetallic compounds is not significant. On the other hand, the solubility of the terminal solid solution phases is not considered due to the lack of available experimental data. It should be pointed out that the calculated phase relations of isothermal sections at different temperatures in Figure 4 are consistent with the experimental results. Of course, we revised carefully the ‘Abstract’ and ‘Results and discussion’ sections in the text. Please check them in the revised manuscript.

Q3. The agreement with prior work was not always as good as the authors stated.

Answer: Thanks for the comment. During the present calculation of the Ag-Cu-Sn ternary system, the available experimental data are taken into account considering the compatible thermodynamic database of the Ag-Cu, Ag-Sn and Cu-sn sub-binary systems. The thermodynamic parameters of the Ag-Cu-Sn ternary system obtained finally can be convenient to extrapolate the thermodynamic database of the multi-component systems. Thus, the present calculations are more suitable than previous calculated results. The related expressions were revised carefully in the text. Please check them in the revised manuscript.

Q4. For ease of clarity, the reference numbers must also be given in the legends of all the diagrams when prior work is plotted for comparison (Figures 2, 4, 5, 6, 7 and 8). This will make the paper much easier to follow. 

Answer: Thanks for the suggestion. We revised the all the figures (Figures 2, 4, 5, 6, 7 and 8) according to the reviewer’s comments. Please check them in the revised manuscript.

Q5. It should be stated in the conclusions that the solid solubilities were not taken into account, although they should be.

Answer: Thanks for the suggestion. We revised the conclusions, which show that the solubility of phases was not taken into account in the present optimization. Please check them in the text.

Q6. More should be stated about the bcc-A2 phase in Cu-Sn. It was stated that a ternary was apparently incorrect, but no further reason was given for stating that the binary was correct.

Answer: Thanks for the comment. In the reported literature, the description of the bcc-A2 and D03 phases are different with each other. In this work, in order to extrapolate conveniently the thermodynamic database of the multi-component systems, the description of the bcc-A2 and D03 phases is simplified as a single phase using the substitutional solution model, which is already accepted in the COST 531 lead-free solders database (Seen in Ref. [4]). The related expressions were revised in the text. Please check them in the revised manuscript.

Q7. The temperatures of some of the curves should be given on Figure 8.

Answer: Thanks for the suggestion. Figure 7 and Figure 8 were revised carefully according to the reviewer’s comments. It was noted that the results of the enthalpies of mixing of liquid alloys calculations at different temperatures are same in Figure 7 because the effect of temperature on the enthalpies of mixing of Ag-Cu-Sn liquid alloys was not considered in the calculations of Moon et al. [21] and Ohnuma et al. [24]. Please check them in the revised manuscript.

Q8. The writing style is long-winded, and the paper should be written in a more succinct way. There are also places where more clarity is needed. These and other corrections are given on the attached scan.

Answer: Thanks for the comments and suggestions. We revised and corrected carefully the manuscript according to these comments. Please check them in the revised manuscript.

Reviewer 2 Report

Great paper!

It is obvious that the authors performed a thorough research.

The paper is definitely ready to be published as is.

Best regards!

Author Response

Reviewer #2

Great paper!

It is obvious that the authors performed a thorough research.

The paper is definitely ready to be published as is.

Best regards!

Answer: Thanks for the comment. There are no any questions to be addressed.

Reviewer 3 Report

I am pleased to state that this is indeed a valuable paper on a important system for lead free solders and brazing alloys as well; generally speaking it does provide a sound basis for further Calphad database development extending  the ternary Ag Cu Sn system.

Using the thermodynamic database search engine:

https://avdwgroup.engin.brown.edu/

it is confirmed  that here are currently a few open databases including this system apart from a paper

https://doi.org/10.1016/j.calphad.2016.01.002

and a paper

which is related to

https://www.msed.nist.gov/phase/solder/NIST-solder.tdb

So this paper deserves to be published as is.

I just suggest two minor variations

1)  Conclusions ( line 309 311)

It indicates that thermodynamic parameters of the Ag-Cu-Sn ternary system obtained in this work would provide to the good basis to develop the compatible thermodynamic database of the multicomponent Ag-Cu-Sn based alloys.

may be it should be changed to

 It indicates that thermodynamic parameters of the Ag-Cu-Sn ternary system obtained in this work  provide a good basis to develop a compatible thermodynamic databases including multicomponent Ag-Cu-Sn based alloys.

2) For the above aim presented in point 1, It would be of great scientific value if the authors would provide for this open journal an open tdb file in the appendix, which would allow the possibility to quickly compare results on different databases like it is possible to do for binary systems in the excellent NIMS collection 

https://cpddb.nims.go.jp/cpddb/periodic.htm

Author Response

Reviewer #3

I am pleased to state that this is indeed a valuable paper on an important system for lead free solders and brazing alloys as well; generally speaking it does provide a sound basis for further Calphad database development extending the ternary Ag Cu Sn system. Using the thermodynamic database search engine: https://avdwgroup.engin.brown.edu/, it is confirmed  that here are currently a few open databases including this system apart from a paper (https://doi.org/10.1016/j.calphad.2016.01.002) and a paper (https://www.metallurgy.nist.gov/phase/papers/JEM(Sn-Ag-Cu).pdf) which is related to https://www.msed.nist.gov/phase/solder/NIST-solder.tdb. So this paper deserves to be published as is. I just suggest two minor variations.

Q1. Conclusions (line 309 311)

It indicates that thermodynamic parameters of the Ag-Cu-Sn ternary system obtained in this work would provide to the good basis to develop the compatible thermodynamic database of the multicomponent Ag-Cu-Sn based alloys.

May be it should be changed to “It indicates that thermodynamic parameters of the Ag-Cu-Sn ternary system obtained in this work provide a good basis to develop a compatible thermodynamic databases including multicomponent Ag-Cu-Sn based alloys.”

Answer: Thanks for the suggestion. We corrected carefully the conclusions in the text based on the reviewer’s comments. Please check them in the text.

Q2. For the above aim presented in point 1, It would be of great scientific value if the authors would provide for this open journal an open tdb file in the appendix, which would allow the possibility to quickly compare results on different databases like it is possible to do for binary systems in the excellent NIMS collection https://cpddb.nims.go.jp/cpddb/periodic.htm

Answer: Thanks for the suggestion. We uploaded the TDB file obtained in this work as supplementary materials, which is useful to produce the calculated results of the Ag-Cu-Sn ternary system. Please check them in the revised manuscript.